# Spike Antibody Titers Evaluation after a 2-Dose Regimen of BNT162b2 Vaccination in Healthcare Workers Previously Infected with SARS-CoV-2

Satoshi Kayukawa,[a] Kengo Nanya,[a] Makoto Morita,[a] Kenji Ina,[b] Yoshihiro Ota,[b] Shinji Hasegawa[a]

[a]Nagoya Memorial Hospital, Nagoya, Japan
[b]Shinseikai Daiichi Hospital, Nagoya, Japan

**KEYWORDS** SARS-CoV-2, antibody titer, past infection, vaccine

The 2-dose BNT162b2 vaccine (Pfizer-BioNTech; reported efficacy 94.8%) regimen against severe acute respiratory syndrome coronavirus 2 (SARS-CoV-2) was authorized in December 2020 (1). However, studies suggest that previously infected individuals can achieve a rapid immune response with a single vaccine dose, compared with SARS-CoV-2-naive individuals (2–5).

To test the effects of one and two vaccinations for individuals with a history of coronavirus disease (COVID-19), volunteers were recruited from among the medical staff at two local hospitals. All participants received two doses of BNT162b2 vaccine with a 3-week interval between doses. Blood samples were taken at days −7 to 0 (baseline), 2 weeks, and 2 months after the first vaccination. A quantitative determination of antibodies against the receptor binding domain of the SARS-CoV-2 S1 subunit of the spike protein was made using plasma samples (Elecsys anti-SARS-CoV-2 S, Roche Diagnostics International Ltd., Rotkreuz, Switzerland). Written informed consent was obtained from participants, and the study was approved by each hospital's ethics committee. Statistical analysis was performed with EZR (6). Differences in antibody titers were evaluated by a $t$ test. A $P$ value of $<0.05$ was considered significant.

Overall, 369 health care workers were enrolled; 22 previously had COVID-19—confirmed by reverse transcriptase PCR (RT-PCR) during the past 3 months—while 347 did not have COVID-19 previously. To minimize confounding, the same number of noninfected individuals (COVID−) were randomly selected from among antibody-negative individuals at baseline to match the age and gender of each infected individual (COVID+). Chronological changes in antibody titers were compared between COVID− and COVID+ groups (Fig. 1). Before vaccination, the baseline antibody titer was 213.6 (139.2 to 288.0) U/mL in the COVID+ group; the COVID− group had no detectable antibodies. Antibody titers increased at 2 weeks in both groups; titers were significantly higher in the COVID+ group than in the COVID− group (11,664 [9,155 to 14,174] U/mL versus 81.8 [21.2 to 142.2] U/mL; $P < 0.001$). However, the titers in the COVID− group at 2 weeks did not reach the baseline levels of the COVID+ group ($P < 0.001$). At 2 months, antibody titers elevated further in the COVID− group ($P < 0.001$) but not in the COVID+ group, although the titer remained higher than that of the COVID− group ($P < 0.001$). No participant developed apparent COVID-19 during the blood sampling period.

Two BNT162b2 vaccination doses were administered within 3 months of COVID-19 outbreak (between December 2020 and February 2021). We found that vaccination increased the antibody titers in SARS-CoV-2-naive individuals to values greater than the baseline levels in COVID+ participants. A further booster effect was not observed

Address correspondence to Satoshi Kayukawa, kayu_kawa@yahoo.co.jp.

**FIG 1** Antibody responses in age- and sex-matched pairs with or without a history of COVID-19. All participants except four (one, anaphylaxis following the first dose; three, quit their job) received two doses of BNT162b2 vaccine; the second dose was administered 18 to 25 days after the first dose was administered. Plasma was drawn before vaccination (day −7 to 0), at 2 weeks (day 15 to 21) after vaccination, and at 2 months (day 57 to 64) after vaccination. Antibody titers against the receptor binding domain of the SARS-CoV-2 S1 subunit of the spike protein were determined using Elecsys anti-SARS-CoV-2 S (Roche Diagnostics International Ltd., Rotkreuz, Switzerland). In cases wherein the detection limit (250 U/mL) was exceeded, plasma samples were diluted 50 to 200 times, as appropriate. To minimize confounding, 22 age- and sex-matched individuals were randomly identified (COVID−) among the uninfected individuals ($n$ = 343), after excluding 4 seropositive individuals. Data of previously infected individuals (COVID+; $n$ = 22) and those of COVID− participants were compared. Both groups showed elevation of antibody titers after the first dose. Following the second dose, titers in COVID− individuals were boosted, but titers for COVID+ individuals did not increase at 2 months. The antibody titers after 2 months in COVID− individuals were higher than the prevaccination antibody titer in COVID+ individuals but lower than the antibody titer at 2 weeks in COVID+ individuals.

in COVID+ participants following the second inoculation. Thus, a single-dose vaccination might be sufficient in recently infected individuals. We also found that antibody titers began to wane at 6 months after the initial series of vaccinations (data not shown); however, this does not necessarily indicate a decrease in neutralizing activity (7). A booster dose of BNT16b2 could have immunogenicity (8), but the timing and recipient prioritization remain controversial (9). Study limitations include small sample size and lack of evidence supporting vaccine efficacy.

We declare no conflicts of interest.

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
