## [Reviewer comments · Microbiology Spectrum]

Microbiology Spectrum

Spike Antibody Titers Evaluation After a 2-Dose Regimen of BNT162b2 Vaccination in Health Care Workers Previously Infected with SARS-CoV-2

Satoshi Kayukawa, Kengo Nanya, Makoto Morita, Kenji Ina, Yoshihiro Ota, and Shinji Hasegawa

Corresponding Author(s): Satoshi Kayukawa, Nagoya Memorial Hospital

Review Timeline:

Submission Date:	July 27, 2021
Editorial Decision:	September 13, 2021
Revision Received:	October 8, 2021
Accepted:	October 24, 2021

Editor: Yongjun Sui

Reviewer(s): The reviewers have opted to remain anonymous.

Transaction Report:

DOI: <https://doi.org/10.1128/Spectrum.01036-21>

September 13, 2021

Dr. Satoshi Kayukawa
Nagoya Memorial Hospital
4-305 Hirabari, Tenpaku-ku
Nagoya 4688520
Japan

Re: Spectrum01036-21 (Spike Antibody Titers Evaluation After a 2-Dose Regimen of BNT162b2 Vaccination in Health Care Workers Previously Infected with SARS-CoV-2)

Dear Dr. Satoshi Kayukawa:

Thank you for submitting your manuscript to Microbiology Spectrum. When submitting the revised version of your paper, please provide (1) point-by-point responses to the issues raised by the reviewers as file type "Response to Reviewers," not in your cover letter, and (2) a PDF file that indicates the changes from the original submission (by highlighting or underlining the changes) as file type "Marked Up Manuscript - For Review Only". Please use this link to submit your revised manuscript - we strongly recommend that you submit your paper within the next 60 days or reach out to me. Detailed information on submitting your revised paper are below.

Link Not Available

Sincerely,

Yongjun Sui

Journals Department
Reviewer comments:

Reviewer #1 (Comments for the Author):

In the manuscript, the authors analyzed the titers of Receptor binding domain of SARS-CoV-2 Spike protein-directed antibody in people with a history of COVID-19 or SARS-CoV-2-naive individuals after one and two doses of BNT162b2 vaccine. Based on the data obtained from a small size of samples, the author found that a two doses of BNT162b2 vaccine increased the antibody titers in SARS-CoV-2-naive individuals, while the second vaccination did not significantly increase the antibody titers in individuals with a history of COVID-19. The author's finding is supported by their clear data, although the data was obtained from a small size of samples as also discussed by the authors. The experiment and methodology of the manuscript are technically sound. The manuscript is suggested to be accepted.

Specific comments are below.

1. The phrase "To verify that two vaccinations....." in line 23 should be changed to "To test the effects of one and two vaccinations....." to match the context of the manuscript.
2. Could the authors discuss whether a third vaccination might further increase the antibody titers in SARS-CoV-2-naive individuals?

Reviewer #2 (Comments for the Author):

The authors determined abs titers only twice: 2 weeks and 2 months after first vaccination. Why the plasma samples were not tested at the same time after each of vaccination doses?

The paper would benefit of a thorough English language correction as there are sentences that are difficult to follow.

Some specific points:

Line 23 - phrase "appropriate" - this term is not clear

Line 36-38 - the sentence is not clear

Line 52 - the authors write that "The time between infection and vaccination has been found to be 1-3 months" - what do they mean by that?

Reviewer #3 (Comments for the Author):

The manuscript is written in standard English but there are a few lines that should be reviewed. Please check lines 25-26 (All participants were planned two doses of....); line 36 ("While" should be deleted); line 53 (should be "individuals were involved in institutional outbreaks).

The author should include any reference/s regarding the approval documents (IRB protocol) by the ethics committee of each hospital if available.

The authors state the U/ml obtained in the study. It should be indicated the linear range of the detection method; for Elecsys anti-SARS-CoV-2 S, I think is 0.4 to 250 U/ml.

The Discussion Section should be expanded. Please, could you discuss the current role of antibody detection and the potential role of neutralizing antibodies in protective immunity?

Staff Comments:

Preparing Revision Guidelines

Please return the manuscript within 60 days; if you cannot complete the modification within this time period, please contact me. If you do not wish to modify the manuscript and prefer to submit it to another journal, please notify me of your decision immediately so that the manuscript may be formally withdrawn from consideration by Microbiology Spectrum.

In the manuscript, the authors analyzed the titers of Receptor binding domain of SARS-CoV-2 Spike protein-directed antibody in people with a history of COVID-19 or SARS-CoV-2-naive individuals after one and two doses of BNT162b2 vaccine. Based on the data obtained from a small size of samples, the author found that a two doses of BNT162b2 vaccine increased the antibody titers in SARS-CoV-2-naive individuals, while the second vaccination did not significantly increase the antibody titers in individuals with a history of COVID-19. The author's finding is supported by their clear data, although the data was obtained from a small size of samples as also discussed by the authors. The experiment and methodology of the manuscript are technically sound.

Specific comments are below.

1. The phrase "To verify that two vaccinations....." in line 23 should be changed to "To test the effects of one and two vaccinations....." to match the context of the manuscript.
2. Could the authors discuss whether a third vaccination might further increase the antibody titers in SARS-CoV-2-naive individuals?

8 October 2021
Dr. Yongjun Sui
Editor, Microbiology Spectrum
Journals Department
Thank you very much for your communication via email on September 13th, 2021. We do appreciate the time and effort you and each of the reviewers have dedicated to provide insightful feedback and suggestions on ways to strengthen our paper entitled, 'Spike Antibody Titers Evaluation After a 2-Dose Regimen of BNT162b2 Vaccination in Healthcare Workers Previously Infected with SARS-CoV-2' (Spectrum01036-21). We have included a point-by-point response to the reviewers' comments. We have highlighted all changes in the revised manuscript. We removed the section headers (Methods, Results, etc.) from my manuscript. We believe that the manuscript has improved following the changes and hope that our manuscript is now suitable for publication in *Microbiology Spectrum*.

Sincerely,

Satoshi Kayukawa, MD, PhD
Nagoya Memorial Hospital
4-305 Hirabari, Tenpaku-ku, Nagoya 468-8520, Japan
Tel: 052-804-1111; Fax: 052-806-5013; E mail: kayu_kawa@yahoo.co.jp

To the reviewers

Reviewer #1

Thank you very much for your kind review of our paper. We revised the manuscript (the revisions are marked by the use of blue-colored font), following your useful and helpful suggestions. In addition, we have kept

track changes on to show any deletions or insertions for minor grammatical changes and word-count reductions.

1. The phrase "To verify that two vaccinations....." in line 23 should be changed to "To test the effects of one and two vaccinations....." to match the context of the manuscript.

Reply: We have changed the expression of line 17 (line 23 before revision). To test the effects of one and two vaccinations for people with a history of coronavirus disease (COVID-19),

2. Could the authors discuss whether a third vaccination might further increase the antibody titers in SARS-CoV-2-naïve individuals?

Reply: Vaccine effectiveness against SARS-CoV-2 infection declines over time among frontline workers including healthcare personnel (CDC. ACIP: August 30, 2021, Meeting). We have preliminary data on the longitudinal monitoring of anti-spike antibody (S-Ab) titers after vaccination in both COVID+ and COVID- groups, as shown in the attached file entitled, "Abs 6-month after vaccination". The levels of S-Ab measured by Roche Elecsys anti-SARS-CoV-2-S begin to wane but are still maintained at 6 months after the initial vaccination. A booster dose of BNT16b2 could have immunogenicity; however, the optimal timing of booster dose administration for vaccinated individuals is still controversial. In addition to humoral responses, cell-mediated immunity provides immunological memory (N Engl J Med. doi: 10.1056/NEJMc2113468. Online ahead of print). Furthermore, in countries where vaccine supply may be insufficient, priority should be given to the unvaccinated; if supply was adequate, the selection of individuals to receive the booster must be prioritized after considering the balance between efficacy and adverse reactions (Lancet doi.org /10.1016/S0140-6736(21)02046-8).

Therefore, we revised the Discussion section as follows:

Thus, a single-dose vaccination might be sufficient in recently infected individuals. We also found that antibody titers began to wane at 6 months after the initial series of vaccinations (data not shown); however, this does

not necessarily indicate a decrease in neutralizing activity (7). A booster dose of BNT16b2 could have immunogenicity (8), but the timing and recipient prioritization remain controversial (9).

We also added references #7, 8 and 9.

7. Bradley BT, Bryan A, Fink SL, Goeker EA, Roychoudhury P, Huang ML, Zhu H, Chaudhary A, Madarampalli B, Lu JYC, Strand K, Whimbey E, Bryson-Cahn C, Schippers A, Mani NS, Pepper G, Jerome KR, Morishima C, Coombs RW, Wener M, Cohen S, Greninger AL. 2021. Anti-SARS-CoV-2 assay are concordant with previously available serologic assays but are not fully predictive of sterilizing immunity. *J Clin Microbiol* 59(9): e0098921. doi: 10.1128/JCM.00989-21. Epub 2021 Aug 18
8. Falsey AR, Frencck RW Jr, Walsh EE, Kitchin N, Absalon J, Gurtman A, Lockhart S, Bailey R, Swanson KA, Xu X, Koury K, Kalina W, Cooper D, Zou J, Xie X, Xia H, Türeci Ö, Lagkadinou E, Tompkins KR, Shi PY, Jansen KU, Şahin U, Dormitzer PR, Gruber WC. 2021. SARS-CoV-2 Neutralization with BNT162b2 Vaccine Dose 3. *N Engl J Med* doi: 10.1056/NEJMc2113468. Online ahead of print.
9. Kraus PR, Fleming TR, Peto R, Longini IM, Figueroa JP, Sterne JA, Cravioto A, Rees H, Higgins JP, Boutron I, Pan H, Gruber MF, Arora N, Kazi F, Gaspar R, Swaminathan S, Ryan MJ, Henao-Restrepo A. 2021. Considerations in boosting COVID-19 vaccine immune responses. *Lancet* doi.org/10.1016/S0140-6736(21)02046-8.

Reviewer #2

The authors determined abs titers only twice: 2 weeks and 2 months after first vaccination. Why the plasma samples were not tested at the same time after each of vaccination doses?

Reply: Since IgG is generally produced 2 weeks after antigen exposure, the initial reaction was confirmed at 2 weeks after the first vaccination.

Concerning the timing of the second measurement, our purpose was to determine the final immunity established by the 2-dose vaccination, but not to compare the degree of first and second reactions. We therefore determined the antibody titers at 2 months after the first vaccination.

The paper would benefit of a thorough English language correction as there are sentences that are difficult to follow.

Reply: Thank you for your kind suggestion. A thorough English language review of the paper was completed by native speakers.

Some specific points:

Line 23 - phrase "appropriate" - this term is not clear

Reply: We agree that this term was not clear and have changed this sentence to:

To test the effects of one and two vaccinations for people with a history of coronavirus disease (COVID-19),

Line 36-38 - the sentence is not clear

Reply: We agree that this sentence was not clear and have made changes. It now reads:

Overall, 369 healthcare workers were enrolled; 22 previously had COVID-19—confirmed by reverse transcriptase-PCR (RT-PCR) during the past 3 months—while 347 did not have COVID-19 previously.

Line 52 - the authors write that "The time between infection and vaccination has been found to be 1-3 months" - what do they mean by that?

Reply: We understand that this is confusing and have made changes to reflect this:

Two BNT162b2 vaccination doses were administered within 3 months of COVID-19 outbreak (between December 2020 and February 2021).

.

Reviewer #3

The manuscript is written in standard English but there are a few lines that should be reviewed. Please check lines 25-26 (All participants were planned

two doses of...); line 36 ("While" should be deleted); line 53 (should be "individuals were involved in institutional outbreaks).

Reply: Thank you very much for your kind review of our paper. We changed the original text (marked by the use of blue font), according to the advice of native English speakers.

All participants ~~received~~~~were planned~~ two doses of BNT162b2 vaccine with a 3-week interval between the doses.

Overall, 369 ~~healthcare workers~~ were enrolled; 22 previously had COVID-19—confirmed by reverse transcriptase-PCR (RT-PCR) during the past 3 months—~~while~~ 347 did not have ~~COVID-19~~ previously.

~~Two BNT162b2 vaccination doses were administered within 3 months of COVID-19 outbreak (between December 2020 and February 2021).~~

~~The author should include any reference/s regarding the approval documents (IRB protocol) by the ethics committee of each hospital if available.~~

Reply: We submitted a copy of the approval documents by the ethics committee of each hospital in Japanese. Please refer to the attached files, "Approval Documents 1, 2".

~~The authors state the U/ml obtained in the study. It should be indicated the linear range of the detection method; for Elecsys anti-SARS-CoV-2 S, I think is 0.4 to 250 U/ml.~~

Reply: Thank you for your important suggestion. The samples were diluted 50–200 times and measured within the detection range. We added the following text in the figure legend to explain precisely the method as follows:

~~Antibody titers against the receptor binding domain of the SARS-CoV-2 S1 subunit of the spike protein were determined using Elecsys anti-SARS-CoV-2 S (Roche Diagnostics International Ltd., Rotkreuz, Switzerland). In cases wherein the detection limit (250 U/mL) was exceeded,~~

plasma samples were diluted 50-200 times, as appropriate

The Discussion Section should be expanded. Please, could you discuss the current role of antibody detection and the potential role of neutralizing antibodies in protective immunity?

Reply: Thank you for your suggestion. The correlation between the chronological changes in antibody titers and neutralizing potency is an important issue in considering protective immunity against viral infection. Just recently it was reported that anti-spike antibody (S-Ab) levels determined by Roche Elecsys anti-SARS-CoV-2-S assay were concordant with anti-spike protein AdviseDxSARS-CoV-2 IgG assay (Abbott) in patients with a history of PCR-confirmed SARS-CoV-2 infection. There is also a high degree of correlation between the AdviseDx and the FDA-authorized surrogate neutralization assay (GeneScript). However, neutralizing activity was maintained even after S-Ab reduction in patients after vaccination (J Clin Microbiol. 2021 Aug 18; 59(9): e0098921. doi: 10.1128/JCM.00989-21. Epub 2021 Aug 18). This implies that the detection of S-Ab should suggest the presence of neutralizing activity, but a decrease in S-Ab does not necessarily mean a decrease in neutralizing activity.

Therefore, we added the following sentence into the Discussion section:

Thus, a single-dose vaccination might be sufficient in recently infected individuals. We also found that antibody titers began to wane at 6 months after the initial series of vaccination (data not shown); however, this does not necessarily mean a decrease in neutralizing activity (7). A booster dose of BNT16b2 could have immunogenicity (8), but controversies surround timing and recipient prioritization (9).

We also added references #7, 8 and 9.

7. Bradley BT, Bryan A, Fink SL, Goeker EA, Roychoudhury P, Huang ML, Zhu H, Chaudhary A, Madarampalli B, Lu JYC, Strand K, Whimbey E, Bryson-Cahn C, Schippers A, Mani NS, Pepper G, Jerome KR, Morishima C, Coombs RW, Wener M, Cohen S, Greninger AL. 2021. Anti-SARS-CoV-2 assay are concordant with previously

available serologic assays but are not fully predictive of sterilizing immunity. *J Clin Microbiol* 59(9): e0098921. doi:

10.1128/JCM.00989-21. Epub 2021 Aug 18

8. Falsey AR, Fenwick RW Jr, Walsh EE, Kitchin N, Absalon J, Gurtman A, Lockhart S, Bailey R, Swanson KA, Xu X, Koury K, Kalina W, Cooper D, Zou J, Xie X, Xia H, Türeci Ö, Lagkadinou E, Tompkins KR, Shi PY, Jansen KU, Şahin U, Dormitzer PR, Gruber WC. 2021. SARS-CoV-2 Neutralization with BNT162b2 Vaccine Dose 3. *N Engl J Med* doi: 10.1056/NEJMc2113468. Online ahead of print.
9. Kraus PR, Fleming TR, Peto R, Longini IM, Figueroa JP, Sterne JA, Cravioto A, Rees H, Higgins JP, Boutron I, Pan H, Gruber MF, Arora N, Kazi F, Gaspar R, Swaminathan S, Ryan MJ, Henao-Restrepo A. 2021. Considerations in boosting COVID-19 vaccine immune responses. *Lancet* doi.org/10.1016/S0140-6736(21)02046-8.

October 24, 2021

Dr. Satoshi Kayukawa
Nagoya Memorial Hospital
4-305 Hirabari, Tenpaku-ku
Nagoya 4688520
Japan

Re: Spectrum01036-21R1 (Spike Antibody Titers Evaluation After a 2-Dose Regimen of BNT162b2 Vaccination in Health Care Workers Previously Infected with SARS-CoV-2)

Dear Dr. Satoshi Kayukawa:

Your manuscript has been accepted, and I am forwarding it to the ASM Journals Department for publication. You will be notified when your proofs are ready to be viewed.

Sincerely,

Yongjun Sui
Editor, Microbiology Spectrum

Journals Department
The manuscript "Spike Antibody Titers Evaluation After a 2-Dose Regimen of BNT162b2 Vaccination in Health Care Workers Previously Infected with SARS-CoV-2" is an interesting work with correct methodology. In initial submission the manuscript was partially written in poor English and had some unclear sentences and vague phrases. As I can see in the presented resubmission, all of my comments have been addressed by the authors and the manuscript has been improved. The language of the manuscript is better and the content is clear. Also, the authors explained scientific rationale of selecting certain time-points of immune response analysis. I have no further comments on the article.